# Improved Self-Supervised Deep Image Denoising

**Samuli Laine**
NVIDIA
slaine@nvidia.com

**Jaakko Lehtinen**
NVIDIA, Aalto University
jlehtinen@nvidia.com

**Timo Aila**
NVIDIA
taila@nvidia.com

## Abstract

We describe techniques for training high-quality image denoising models that require only single instances of corrupted images as training data. Inspired by a recent technique that removes the need for supervision through image pairs by employing networks with a "blind spot" in the receptive field, we address two of its shortcomings: inefficient training and poor final denoising performance. This is achieved through a novel blind-spot convolutional network architecture that allows efficient self-supervised training, as well as application of Bayesian distribution prediction on output colors. Together, they bring the self-supervised model on par with fully supervised deep learning techniques in terms of both quality and training speed in the case of i.i.d. Gaussian noise.

## 1 Introduction

Denoising, the removal of noise from images, is a major application of deep learning. Several architectures have been proposed for general-purpose image restoration tasks, e.g., U-Nets (Ronneberger et al., 2015), hierarchical residual networks (Mao et al., 2016), and residual dense networks (Zhang et al., 2018). Traditionally, the models are trained in a supervised fashion with corrupted images as inputs and clean images as targets, so that the network learns to remove the corruption.

Lehtinen et al. (2018) introduced Noise2Noise training, where pairs of corrupted images are used as training data. They observe that when certain statistical conditions are met, a network faced with the impossible task of mapping corrupted images to corrupted images learns, loosely speaking, to output the "average" image. For a large class of image corruptions, the clean image is a simple per-pixel statistic — such as mean, median, or mode — over the stochastic corruption process, and hence the restoration model can be supervised using corrupted data by choosing the appropriate loss function to recover the statistic of interest.

While removing the need for clean training images, Noise2Noise training still requires at least two independent realizations of the corruption for each training image. While this eases data collection significantly compared to noisy-clean pairs, large collections of (single) poor images are still much more widespread. This motivates investigation of self-supervised training: how much can we learn from just looking at bad data? While foregoing supervision would lead to the expectation of some regression in performance, can we make up for it by making stronger assumptions about the corruption process? In this paper, we show that under the assumption of additive Gaussian noise that is i.i.d. between pixels, no concessions in denoising performance are necessary.

We draw inspiration from the recent Noise2Void (N2V) training technique of Krull et al. (2018). The algorithm needs no image pairs, and uses just individual noisy images as training data, assuming that the corruption is zero-mean and independent between pixels. The method is based on *blind-spot networks* where the receptive field of the network does not include the center pixel. This allows using the same noisy image as both training input and training target — because the network cannot see the correct answer, using the same image as target is equivalent to using a different noisy realization. This approach is self-supervised in the sense that the surrounding context is used to predict the value of the output pixel without a separate reference image (Doersch et al., 2015).

The networks used by Krull et al. (2018) do not have a blind spot by design, but are trained to ignore the center pixel using a masking scheme where only a few output pixels can contribute to the loss function, reducing training efficiency considerably. We remedy this with a novel architecture that allows efficient training without masking. Furthermore, the existence of the blind spot leads to poor denoising quality. We derive a scheme for combining the network output with data in the blind spot, bringing the denoising quality on par with conventionally trained networks.

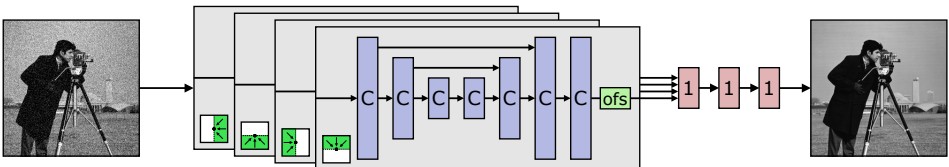

Figure 1: In our blind-spot network architecture, we effectively construct four denoiser network branches, each having its receptive field restricted to a different direction. A single-pixel offset at the end of each branch separates the receptive field from the center pixel. The results are then combined by 1×1 convolutions. In practice, we run four rotated versions of each input image through a single receptive field -restricted branch, yielding a simpler architecture that performs the same function. This also implicitly shares the convolution kernels between the branches and thus avoids the four-fold increase in the number of trainable weights.

## 2 CONVOLUTIONAL BLIND-SPOT NETWORK ARCHITECTURES

Our convolutional blind-spot networks are designed by combining multiple branches that each have their receptive field restricted to a half-plane (Figure 1) that does not contain the center pixel. The principle of limiting the receptive field has been used in PixelCNN (van den Oord et al., 2016a) image synthesis networks, where only pixels synthesized before the current pixel are allowed in the receptive field. We combine the four branches with a series of 1×1 convolutions to obtain a receptive field that can extend arbitrarily far in every direction but does not contain the center pixel.

In order to transform a restoration network into one with a restricted receptive field, we modify each individual layer so that its receptive field is fully contained within one half-plane, including the center row/column. The receptive field of the resulting network includes the center pixel, so we offset the feature maps by one pixel before combining them. Layers that do not extend the receptive field, e.g., concatenation, summation, 1×1 convolution, etc., can be used without modifications.

**Convolution layers.** To restrict the receptive field of a zero-padding convolution layer to extend only, say, upwards, the easiest solution is to offset the feature maps downwards when performing the convolution operation. For an $h \times w$ kernel size, a downwards offset of $k = \lfloor h/2 \rfloor$ pixels is equivalent to using a kernel that is shifted upwards so that all weights below the center line are zero. Specifically, we first append $k$ rows of zeros to the top of input tensor, then perform the convolution, and finally crop out the $k$ bottom rows of the output.

**Downsampling and upsampling layers.** Many image restoration networks involve downsampling and upsampling layers, and by default, these extend the receptive field in all directions. Consider, e.g., a bilinear $2 \times 2$ downsampling step followed immediately by a nearest-neighbor $2 \times 2$ upsampling step. The contents of every $2 \times 2$ pixel block in the output now correspond to the average of this block in the input, i.e., information has been transferred in every direction within the block. We fix this problem by again applying an offset to the data. It is sufficient to restrict the receptive field for the *pair* of downsampling and upsampling layers, which means that only one of the layers needs to be modified, and we have chosen to attach the offsets to the downsampling layers. For a $2 \times 2$ bilinear downsampling layer, we can restrict the receptive field to extend upwards only by padding the input tensor with one row of zeros at top and cropping out the bottom row before performing the actual downsampling operation.

## 3 BAYESIAN TRAINING AND MAP DENOISING

In their basic form, blind-spot networks suffer from the inability to utilize the data at the center pixel at test time; yet, clearly, the observed value carries information about the underlying clean signal. For training it is mandatory to disconnect the information flow from pixel position to itself, but there is no such restriction when using the network to restore novel images after it has been trained. We capitalize on this by training the network to predict, based on the context, a distribution of values instead of a single mean prediction, and applying maximum a posteriori estimation at test time.

In Bayesian training (Nix & Weigend, 1994; Le et al., 2005; Kendall & Gal, 2017), the network predicts output distributions using a negative log-likelihood loss function. We model the data using multivariate Gaussian distributions: For images with $c$ color components, we have the denoising network output a vector of means $\boldsymbol{\mu}_y$ and a covariance matrix $\boldsymbol{\Sigma}_y$ for each pixel. For convenience,

we parameterize the $c \times c$ inverse per-pixel covariance matrix as $\boldsymbol{\Sigma}_y^{-1} = \mathbf{A}_y^{\mathrm{T}} \mathbf{A}_y$, where $\mathbf{A}_y$ is an upper triangular matrix. This choice ensures that $\boldsymbol{\Sigma}_y$ is positive semidefinite with non-negative diagonal entries, as required for a covariance matrix. For RGB images, the network thus outputs a total of nine values per pixel: the three-component mean $\boldsymbol{\mu}_y$ and the six nonzero elements of $\mathbf{A}_y$.

Let $f(\boldsymbol{y}; \boldsymbol{\mu}_y, \boldsymbol{\Sigma}_y)$ denote the probability density of a multivariate Gaussian distribution $\mathcal{N}(\boldsymbol{\mu}_y, \boldsymbol{\Sigma}_y)$ at target pixel color $\boldsymbol{y}$, i.e., $\exp[-\frac{1}{2}(\boldsymbol{y} - \boldsymbol{\mu}_y)^{\mathrm{T}} \boldsymbol{\Sigma}_y^{-1}(\boldsymbol{y} - \boldsymbol{\mu}_y)]/\sqrt{(2\pi)^c |\boldsymbol{\Sigma}_y|}$. Under our parameterization, the corresponding negative log-likelihood loss to optimize during training is

$$loss(\boldsymbol{y}, \boldsymbol{\mu}_y, \mathbf{A}_y) = -\log f(\boldsymbol{y}; \boldsymbol{\mu}_y, (\mathbf{A}_y^{\mathrm{T}} \mathbf{A}_y)^{-1}) = \tfrac{1}{2}||\mathbf{A}_y(\boldsymbol{y} - \boldsymbol{\mu}_y)||^2 - \log|\mathbf{A}_y| + C, \quad (1)$$

where $C$ is a constant term that can be discarded. Because $\mathbf{A}_y$ is a triangular matrix, its determinant $|\mathbf{A}_y|$ is the product of its diagonal elements. To avoid numerical issues, we clamp this determinant to a small positive epsilon ($\epsilon = 10^{-8}$) so that the logarithm is always well-defined.

We assume that all of our images are corrupted by additive uniform Gaussian noise $\mathcal{N}(0, \sigma^2 \mathbf{I})$ with a known standard deviation $\sigma$. Using noisy targets means that there is a "baseline" level of noise in the network output distributions that we must discount. Thanks to the blind spot, the network output is independent of the noise in the center pixel, so their (co-)variances are additive. We can therefore calculate $\boldsymbol{\Sigma}_p = \boldsymbol{\Sigma}_y - \sigma^2 \mathbf{I}$ to determine the actual uncertainty $\boldsymbol{\Sigma}_p$ of the network. To avoid negative variances due to approximation errors, the diagonal elements of $\boldsymbol{\Sigma}_p$ can be clamped to zero.

Let us now derive our maximum a posteriori (MAP) denoising procedure. For each pixel, our goal is to find the most likely clean value $\hat{\boldsymbol{x}}$ given our knowledge of the noisy value $\tilde{\boldsymbol{x}}$ and the output distribution predicted by the network based on the blind-spot neighborhood $\Omega$. It follows that

$$\hat{\boldsymbol{x}} = \underset{\boldsymbol{x}}{\mathrm{argmax}}\, P(\boldsymbol{x}|\tilde{\boldsymbol{x}}, \Omega) = \underset{\boldsymbol{x}}{\mathrm{argmax}}\, P(\tilde{\boldsymbol{x}}|\boldsymbol{x})P(\boldsymbol{x}|\Omega) = \underset{\boldsymbol{x}}{\mathrm{argmax}}\, f(\boldsymbol{x}; \tilde{\boldsymbol{x}}, \sigma^2 \mathbf{I})\, f(\boldsymbol{x}; \boldsymbol{\mu}_y, \boldsymbol{\Sigma}_p), \quad (2)$$

where we have first applied Bayes' theorem to obtain the MAP objective $P(\tilde{\boldsymbol{x}}|\boldsymbol{x})P(\boldsymbol{x}|\Omega)$, and then expressed the associated probabilities as pdfs of Gaussian distributions. In the first term we have exploited the symmetry of the Gaussian distribution, and as the prior term $P(\boldsymbol{x}|\Omega)$ we use the prediction of the network with the baseline uncertainty removed. Following Bromiley (2003), the mean, and consequently the $\mathrm{argmax}$, of this product of two Gaussian distributions is

$$\hat{\boldsymbol{x}} = (\boldsymbol{\Sigma}_p^{-1} + \sigma^{-2} \mathbf{I})^{-1}(\boldsymbol{\Sigma}_p^{-1} \boldsymbol{\mu}_y + \sigma^{-2} \tilde{\boldsymbol{x}}). \quad (3)$$

## 4 RESULTS AND CONCLUSIONS

For the baseline experiments, as well as for the backbone of our blind-spot networks, we use the same U-Net (Ronneberger et al., 2015) architecture as Lehtinen et al. (2018), see their appendix for details. The only differences are that we have layers DEC_CONV1A and DEC_CONV1B output 96 feature maps like the other convolution layers at the decoder stage, and layer DEC_CONV1C is removed. After combining the four receptive field restricted branches, we thus have 384 feature maps. These are fed into three successive $1 \times 1$ convolutions with 384, 96, and $n$ output channels, respectively, where $n$ is the number of output components for the network. All convolution layers except the last $1 \times 1$ convolution use leaky ReLU with $\alpha = 0.1$ (Maas et al., 2013). All networks were trained using Adam with default parameters (Kingma & Ba, 2015), learning rate $\lambda = 0.0003$, and minibatch size of 4. As training data, we used random $256 \times 256$ crops from the 50K images in the ILSVRC2012 (Imagenet) validation set. The training continued until 1.2M images were shown to the network. All training and test images were corrupted with Gaussian noise, $\sigma = 25$.

Table 1 shows the denoising quality in dB for the four test datasets used. From the BSD300 dataset we use the 100 validation images only. Similar to Krull et al. (2018), we use the grayscale version of the BSD68 dataset — for this case we train a single-channel ($c = 1$) denoiser using only the luminance channel of the training images. All our blind-spot noise-to-noise networks use the convolutional architecture (Section 2) and are trained without masking. In BSD68 our simplified *L2* variant closely matches the original NOISE2VOID training, suggesting that our network with an architecturally enforced blind spot is approximately as capable as the masking-based network trained by Krull et al. (2018). We see that the denoising quality of our *Full* setup (Section 3) is on par with baseline results of N2N and N2C, and clearly surpasses standard blind-spot denoising (*L2*) that does not exploit the information in the blind spot. Doing the estimation separately for each color

| Test set | Previous work | | Blind-spot noise-to-noise (our) | | | Baseline | |
|---|---|---|---|---|---|---|---|
| | BM3D | N2V | Full | Per-comp. | L2 | N2N | N2C |
| KODAK | 31.82 | – | **32.39** | 31.54 | 30.58 | **32.39** | **32.41** |
| BSD300 | 30.34 | – | **30.99** | 29.87 | 28.61 | **31.03** | **31.04** |
| BSD68 (grayscale) | 28.59 | 27.71 | **29.27** | **29.27** | 27.76 | **29.30** | **29.31** |
| SET14 | 30.50 | – | **31.20** | 30.54 | 29.51 | **31.17** | **31.17** |
| Average | 30.31 | – | **30.96** | 30.31 | 29.11 | **30.97** | **30.98** |

Table 1: PSNR for various methods and test sets. Numbers for *BM3D* and *N2V* are from (Lehtinen et al., 2018) and (Krull et al., 2018). *Full* is our blind-spot training and denoising method as described in Section 3. *Per-comp.* is an ablated setup where each color component is treated as an independent univariate Gaussian, highlighting the importance of expressing color outputs as multivariate distributions. *L2* refers to training using the standard L2 loss function and ignoring the center pixel when denoising. Columns *N2N* and *N2C* refer to NOISE2NOISE training of Lehtinen et al. (2018) and traditional supervised training with clean targets (i.e., noise-to-clean), respectively. Results within 0.05 dB of the best result for each dataset are shown in boldface.

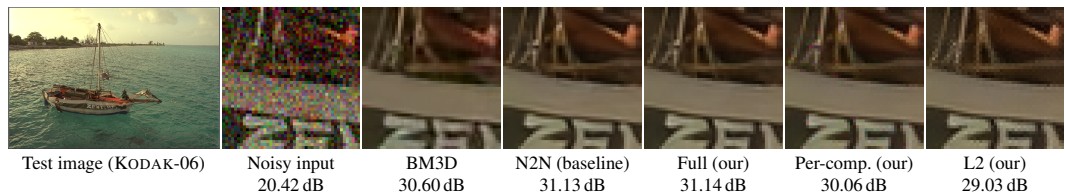

| Test image (KODAK-06) | Noisy input
20.42 dB | BM3D
30.60 dB | N2N (baseline)
31.13 dB | Full (our)
31.14 dB | Per-comp. (our)
30.06 dB | L2 (our)
29.03 dB |

Figure 2: Image quality examples of the various denoising methods. PSNRs are for the single test image.

channel (*Per-comp.*) performs significantly worse, except in the grayscale BSD68 dataset where it is equivalent to the *Full* method.

Figure 2 shows example denoising results. Our *Full* setup produces images that are virtually identical to the N2N baseline both visually and in terms of PSNR. The ablated *Per-comp.* setup tends to produce color artifacts, demonstrating the shortcomings of the simpler per-component univariate model. Finally, the *L2* variant that ignores the center pixel during denoising produces visible checkerboard patterns, some of which can also be seen in the result images of Krull et al. (2018).

**Conclusions.** We have shown that self-supervised training — looking at noisy images only, without the benefit of seeing the same image under different noise realizations — is sufficient for learning deep denoising models on par with those that make use of another realization as a training target, be it clean or corrupted. Currently this comes at the cost of assuming pixel-wise independent noise with a known analytic likelihood model.

## 5 RELATED WORK

PixelCNNs (van den Oord et al., 2016b;a; Salimans et al., 2017) generate novel images in a scanline order, one pixel at a time, by conditioning the possible pixel colors using all previous, already generated pixels. The training uses masked convolutions that prevent looking at pixels that would not have been generated yet — one good implementation of masking (van den Oord et al., 2016a) combines a vertical half-space (previous scanlines) with a horizontal line (current scanline). In our application we use four half-spaces to exclude the center pixel only. Regrettably the term "blind spot" has a slightly different meaning in PixelCNNs: van den Oord et al. (2016a) uses it to denote valid input pixels that the network in question fails to see due to poor design, whereas we follow the naming convention of Krull et al. (2018) so that a blind spot is always intentional.

Applying Bayesian statistics to denoising has a long history. Non-local means (Buades et al., 2005), BM3D (Dabov et al., 2007), and WNNM (Gu et al., 2014) identify a group of similar pixel neighborhoods and estimate the center pixel's color from those. This is conceptually similar to our solution, which uses a convolutional network to represent the mapping from neighborhoods to the distilled outputs. Both approaches need only the noisy images, but while the explicit block-based methods determine a small number of neighborhoods from the input image alone, our blind-spot training can implicitly identify and regress an arbitrarily large number of neighborhoods from a collection of noisy training data.

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
