# OpenReview forum: "Improved Self-Supervised Deep Image Denoising"
_ICLR.cc/2019/Workshop/LLD — LLD 2019_

### Official Review · AnonReviewer2 · 2019-04-07
**Interesting method that is well-described; additional experiments could strengthen the paper**

**Rating:** 4
**Confidence:** 2

**Review:**

Paper Summary:
The authors present a method by which a denoising network can be trained using only examples of single noisy images.  The technique relies on idea of blind-spot networks, where the center pixel is eliminated from the receptive field, and reasonably strong statistical assumptions about the noise profile.  The authors define a network architecture based on a directional CNN applied to four rotations of the same image, with the receptive field of each restricted to a half-plane (just excluding the center row/column), which is then combined via a set of 1-D convolutions.  The network is then optimized to learn the parameters of a multivariate Gaussian distribution describing the mean and covariance matrix characterizing each pixel value.  Using the assumption of additive Gaussian noise that is IID between the pixels, these parameters are used in a MAP estimation procedure at test time to arrive at the denoised value.  Empirical performance on several datasets appears promising.

Quality (Pros):
The technical content of the paper represents an interesting approach to the denoising problem, which draws heavy inspiration from the recent Noise2Void technique.  Proposed improvements over this previous technique include (1) a more efficiently trainable network architecture (2) utilization of a Bayesian modeling approach and (3) integration of information from the center pixel at test time.  The network architecture itself is adequately described in Section 2, and the design choices seem reasonable.  Given the length requirements of the paper, it is reasonable that ablations were not performed on network design, but the lack of quantitative training efficiency comparisons to the Noise2Void technique limits this aspect of the contribution.  The Bayesian training approach seems a natural fit for this type of denoising problem, where the noise in an image is characterized by a probability distribution that is local to the pixel in question; of course, the assumption the the noise is additive, Gaussian, and IID amongst pixels is strong, but the relatively straightforward MAP estimation procedure these assumptions enable is a strength of the proposed approach.  Finally, integration of integration from the center pixel at test time, but not train time, is an important result of this modeling approach.

Overall, the paper presents an interesting technique that contains several distinct improvements over previous approaches, and acknowledges that these are applicable only under strong statistical assumptions.  The performance experiments appear well-posed to assess how the proposed technique compares to appropriate baselines (Noise2Void, Noise2Noise, Noise2Clean), but comparison to the most similar baseline (Noise2Void) is only presented for one dataset.  Though these results appear compelling, the paper could be improved by adding the Noise2Void numbers for other tested datasets.

Clarity
The paper is clearly written, and at an appropriate level of detail.

Significance
In addition to suggesting the potential for improved image denoising results using only single noisy images for training, this paper presents an interesting application of Bayesian modeling approaches for denoising that may inspire future work that yields similar results without the strong statistical assumptions currently imposed on the noise distribution.

Limitations (Cons)
Current limitations of this work are as follows:
(1) Lack of experiments quantitatively demonstrating training efficiency relative to Noise2Void
(2) Lack of experiments demonstrating comparison to Noise2Void on any dataset but BSD68
(3) No description of the various datasets in Table 1 is given; this would be helpful
(4) Strong statistical assumptions required on the noise distributions -- to be clear, this is less a limitation of the authors’ analysis, but rather of the setting they chose to analyze; in particular, the assumption that all images are characterized by additive Gaussian noise with the same standard deviation, and that this standard deviation is known, may be limiting in practice
(5) It would be instructive to demonstrate how performance of this method compares to others if statistical assumptions on the noise distribution are violated.

---

### Official Review · AnonReviewer1 · 2019-04-09
**An interesting architecture and training technique for self-supervised denoising**

**Rating:** 4
**Confidence:** 2

**Review:**

The authors propose techniques for training image denoising models without supervision, using only corrupted images and seeing each corrupted image once.

Positives:
- Interesting blind-spot network architecture with "4-directional" reading head and shared weights;
- The Bayesian training is also a nice contribution, and proves to be important in the evaluation;
- Concisely written and straight to the point

Remarks & questions:
a) One extra baseline I am curious of: one could train N2N under the same constraint of using each image once using extra synthetic noise. Given a corrupted image, one can add extra corruption to obtain copies of it with different noise levels, and this could be enough supervision to train N2N to reasonable accuracy. It would be an interesting additional baseline, since it would use the same constraint as your work.
b) Inference speed: can you compare the inference speed between your network and other baselines e.g. N2N?
c) It seems to me that the datasets used in evaluation have standard image artifacts: JPEG, scale, noise... This makes the problematic of designing methods for seeing each image once a bit artificial because N2N is already well suited to the task. I wonder if one could showcase the benefits of the method on datasets with a special type of artifacts, e.g. medical imagery or hyperspectral data.

Overall I think the paper is a worthwhile contribution to the workshop.

---

### Decision · Program_Chairs · 2019-04-09
**Acceptance Decision**

Accept